# Effect of the Increasing Operator's Experience on the Miniscrew Survival Rate

Joanna Lis , Kornelia Rumin, Michał Sarul and Beata Kawala *

Department of Dentofacial Orthopedics and Orthodontics, Wroclaw Medical University, 50-425 Wroclaw, Poland
* Correspondence: beata.kawala@umw.edu.pl; Tel.: +48-71-784-02-99

**Abstract:** The aim of our study was to determine the learning curve of orthodontic miniscrew insertion in novices, unsupervised and supervised by experts. Three inexperienced orthodontists, two unsupervised (JA-S, MS) and one supervised (JŁ) by the experts, inserted 404 AbsoAnchor® miniscrews (Dentos, SH131208) into 202 patients for en masse retraction or distalization purposes. Miniscrews were inserted symmetrically on both sides of the maxilla between the 2nd bicuspid and the 1st molar. Then, the miniscrew survival rate and the orthodontists' learning curves were estimated. After first 40 insertions JA-S, MS, and JŁ achieved 73, 70, and 83% of stable miniscrews, respectively. The overall outcome showed that after 74 and 118 insertions, the miniscrew survival rate reached 80% and 85%, respectively, and from that point the learning curve still presented an ascending trend. The success rate increased significantly: by 1.016 times with every procedure (odds ratio = 1.016, $p = 0.006$). Since the success rate significantly increased after each miniscrew insertion in the buccal maxillary area, this finding might encourage hesitating clinicians to introduce the miniscrews to the daily practices. Obtained data could also help to plan an effective training system for novice orthodontists.

**Keywords:** skeletal anchorage; miniscrews; orthodontic treatment

## 1. Introduction

Temporary anchorage devices (TADs) have become widely popular due to their ability to provide compliance-free skeletal anchorage, which is irreplaceable in some clinical cases. Since every clinician's aim is to advocate the medical procedure perfectly, over the years researches have thoroughly investigated the risk factors associated with TADs failure, such as: age, sex, type of malocclusion, location, thickness of mucosa, type of mucosa, thickness of the cortical bone, position (left or right side), proximity of the tooth roots, hygiene, smoking habits, TAD type, length and diameter, time of loading, applied force, type of movement, and clinician's experience at the moment of TAD insertion [1].

To date, only a few clinical studies have assessed the learning curve of TAD insertion or progressive clinician's experience as variables influencing TAD stability [2–7]. The evaluation of such a learning curve could improve the accuracy of insertion and provide patients with safe and high-quality care. Therefore, the aim of our study was to determine how the operator's experience, increasing over time, affects the TAD survival rate in patients with bilaterally inserted miniscrews in the posterior region of the maxilla. To do so, we formulated two hypotheses:

1. There is a correlation between increasing operator's experience and percentage of failure/success rates,
2. There is a threshold in terms of the number of inserted TADs that secures a relatively high (80% and 85%) and subsequently increasing success rate.

## 2. Materials and Methods

In order to determine effect of the increasing operator's experience on the miniscrew survival rate, the learning curves of two orthodontists JA-S and MS, novices when it came

to implanting TADs, were analyzed and—seven years later—compared with an analogous chart obtained by the third clinician JŁ, a postgraduate student trained and supervised by JA-S and MS.

The analyzed study material comprised 202 generally healthy patients (both genders) aged from 20 to 50 years, without symptoms of oral diseases and with good oral hygiene, treated at the Department of Dentofacial Orthopedics and Orthodontics, Wroclaw Medical University. All patients presented with skeletal Class II, with either a hypodivergent or normal angle between the maxillary and mandibular planes, with excellent oral hygiene determined by the Simplified Oral Hygiene Index (OHI-S), which was rated as good (0.0–1.2). Records also showed that the patients with massive frenula, which could potentially irritate the TAD's heads while chewing or during facial muscles activity, were disqualified as the recipients of skeletal anchorage reinforcement.

In total, patients received 404 TADs. Within first six months JA-S and MS inserted 160 TADs each, whilst JŁ implanted 84 TADs within an analogous period. Miniscrews came from one manufacturer, AbsoAnchor® (Dentos, Daegu, South Korea), and served as an anchorage for symmetrical dental movements: either distalization of the maxillary dentition or en masse retraction of six anterior maxillary teeth after extraction of first premolars. Eight-millimeter long, made of titanium alloy (Ti-6Al-4V), tapered (diameter of 1.3 mm at the neck and 1.2 at the apex) miniscrews SH1312-08 were inserted on both sides of the maxilla between the 2nd bicuspid and the 1st molar, in a high position (nonkeratinized mucosa) for en masse retraction or in a low position (keratinized mucosa) for distalization purposes. All TADs were loaded directly with the NiTi coil-springs (Dentos, Daegu, South Korea), delivering a force of 150 g–200 g.

All clinicians applied the Wroclaw protocol of TAD insertion and management, with mandatory elements of (a) securing perpendicular insight into the operative area; (b) vertical stab incision (3–4 mm) and pre-drilling with a working speed of 500–1000 rpm under massive saline irrigation; (c) 30°–40° angulation of TAD to the alveolar process in order to reduce the risk of the root injury, as well as not to impair the tooth movement; (d) ligature wire extensions in every TAD inserted in a high position; (e) loading two weeks after TAD insertion and the healing of aseptic inflammation; (f) no prescription of analgesics or antibiotics; and (g) post-operative instructions for the patients to maintain perfect oral hygiene, to apply gel with chlorhexidine (Elugel®, Pierre Fabre) around TADs for the first 4 weeks after insertion and to avoid any accidental hits against TADs, e.g., with a toothbrush [8].

Severe mobility requiring replacement, as well as spontaneous TAD loss prior to finishing an orthodontic treatment, counted as their failure; otherwise, they were considered stable.

In the case of each doctor, TAD stability/instability was checked at every control visit till the end of orthodontic treatment, which lasted 16 (±5) months on average.

*Statistical Analysis*

The acquired data was subjected to statistical analysis with STATISTICA (data analysis software system), version 12, StatSoft, Inc. (2014), Tulsa, OK, United States) in order to confirm or deny the hypotheses. In order to determine the best possible model, stepwise regression, which includes forward selection and backwards elimination, was applied. Next, the logistic regression models, relative risk, and odds ratio were calculated for each physician separately and in total. Goodness-of-fit was determined to show how accurately the results supported the decision making process. Statistical significance was set at $p = 0.05$.

## 3. Results

Total success rate reached 79.46% (321/404). Separately, it equaled 80% (128/160) for JA-S, 76.25% (122/160) for MS, and 84.52% (71/84) for JŁ.

The success rate with the following groups of 40 TAD insertions is demonstrated in the Table 1. Evaluating each orthodontist's separately, after the first 40 insertions, JA-S, MS, and JŁ achieved 72.5%, 70%, and 82.5% of stable TADs, respectively. Then, the discussed

percentage increased up to 90%, 87.5%, and 85%. As can be seen on the overall learning curve, 80% and 85% TAD survival rates occurred after 74 and 118 insertions, respectively (Figure 1).

**Table 1.** Success rate of TAD insertions (according to groups of 40 TADs) for each doctor separately and in total.

| | Success Rate % | | | |
|---|---|---|---|---|
| | **First Group of 40 TADs** | **Second Group of 40 TADs** | **Third Group of 40 TADs** | **Fourth Group of 40 TADs** |
| **JAS** | 72.5% | 77.5% | 80% | 90% |
| **MS** | 70% | 67.5% | 80% | 87.5% |
| **JL** | 82.5% | 85% | | |
| **Total** | 75% | 76.7 | 80% | 88.8% |

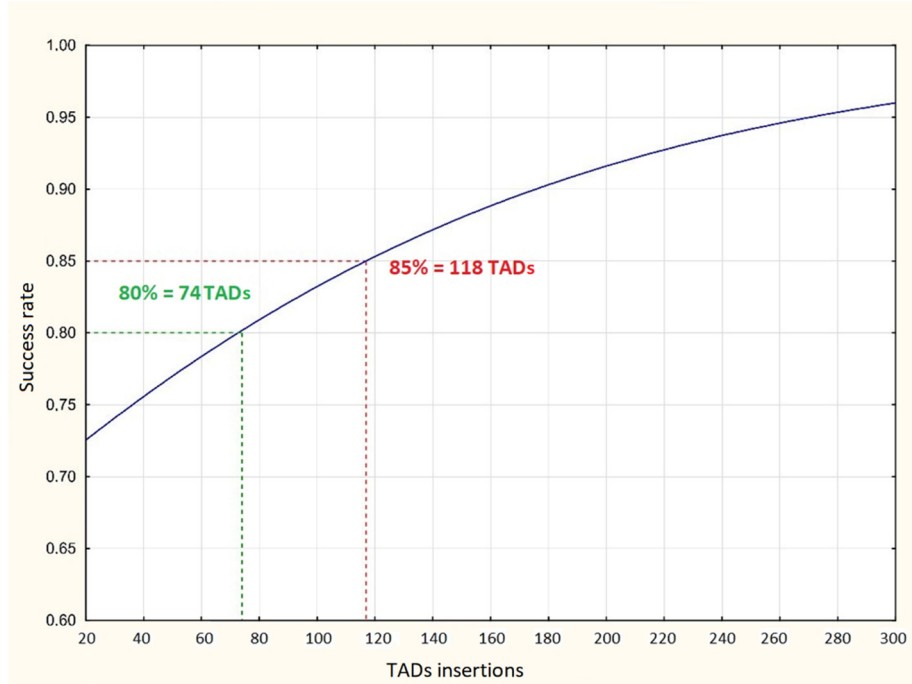

**Figure 1.** Total learning curve in TAD insertion—logistic regression model.

From that point, the TAD total success rate still presented an ascending tendency.

As can be seen from the learning curves, the postgraduate student JŁ exceeded 80% success rate from the start, while for JA-S and MS, the 80% threshold was reached after 76 and 98 TAD insertions, respectively. A success rate of 85% was achieved by JŁ, JA-S, and MS after 48, 112, and 130 TAD insertions, respectively (Figure 2).

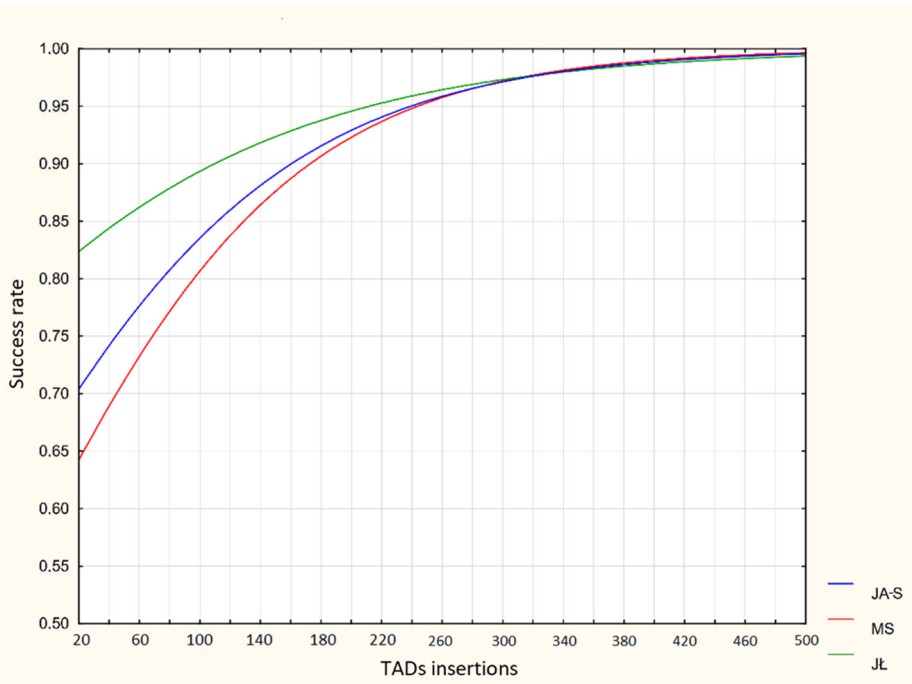

**Figure 2.** Each operator's learning curves in TAD insertion—logistic regression model.

Statistical analysis proved that effectiveness in achieving stability increased by 1.016 ($p$ = 0.006) times with every TAD placement (Table 2), and it was significantly affected by the number of executed procedures or—in other terms—the operator's gradually gathered experience. The chance of a successful procedure did not depend on a particular doctor.

**Table 2.** Results of the logistic regression model.

| | β Coefficient | Standard Error | Upper Limit of 95% CI | Lower Limit of 95% CI | *p* Value | Odds Ratio | Lower Limit of 95% CI for OR | Upper Limit of 95% CI for OR |
|---|---|---|---|---|---|---|---|---|
| Intercept | 0.814 | 0.222 | 0.379 | 1.249 | 0.000 | 2.257 | 1.461 | 3.487 |
| Procedure (TAD insertion) | 0.016 | 0.006 | 0.004 | 0.027 | 0.006 | 1.016 | 1.004 | 1.027 |

## 4. Discussion

Clinician's experience and expertise might significantly affect any medical procedure. Our study proved that this rule also applies to survival rate of TADs, which were implanted symmetrically in the maxillary buccal region. Kim et al. [2] surveyed 210 palatal miniscrews implanted by the same operator within the first 6 years of his experience. After 18 months or the insertion of 36 TADs, the success rate equaled 75% and then gradually improved to over 90%, finally reaching 97.9%. In our study, we achieved similar results at the beginning, although at the end of the survey none of the three clinicians exceeded the 90% success rate. This may be explained by two factors: 1. the lower number of TADs inserted by a single doctor in our research, and 2. the edentulous mid-palatal area utilized by Kim et al. [2], which has already been proven to secure superior TAD stability. This minor risk of TAD loss from the palate is owed to a compact bone and thin gingival tissue in this area, contrary to the buccal aspect utilized in our study. Lim et al. [5], who reported 100% stable mid-palatal miniscrews (25/25), although assessed prior to orthodontic force application, also confirmed such superiority of palate for the TAD location. In the study by Lim et al. [5], one operator also inserted 353 miniscrews in the sites other than the midpalatal one (the maxillary buccal molar, the palatal slope, the mandibular buccal molar, and the mandibular buccal canine area). Regardless of the fact that the stability increased after the insertion of

40 TADs, the authors eventually determined that the clinician's experience did not affect the success rate in statistically significant manner.

Although many studies have reported the success of TADs placed by the same operator, it is still interesting to compare the TAD progressive survival rate for a beginner and a well experienced professional, something that is absent from the literature. Oh et al. [9], who evaluated the success rates achieved by clinicians with various levels of experience, found the score of the professor (98.1%) remarkably higher than that of the postgraduate students (70.8%). The authors concluded that among many factors affecting the success of miniscrews, the operator's skill must not be neglected. Our project not only proved the role of experience in achieving the TAD high success rate but also demonstrated the process of the learning curve itself. Our current experts (JA-S and MS) provided the data collected at the beginning of their involvement in TAD insertion, when JA-S was an associate professor and MS a lecturer, which explains the relatively poor results after the first 40 insertions. However, the youngest doctor (JŁ) who received his training from by that time already experienced orthodontists—JA-S was already a professor and MS an associate professor—demonstrated fast-ascending learning curve. It fully justifies the statement that the learning of miniscrew insertion is highly beneficial and efficient, if supported by the expertise of more knowledgeable colleagues. However, according to the results of the logistic regression analysis, the chance of a successful procedure did not depend on the doctor (their academical or clinical experience) but only on the number of procedures performed when learning a new technique. Although increasing the probability of the miniscrew success by 1.016 times with each procedure may not seem like much, with any learning curve consistency is the key. The role of growing experience, remarkably contributing to the higher success rate of TADs, was also confirmed by the study of Fritz et al. [10], which investigated the success rate of the miniscrews serving as anchorage for premolar distalization and molar uprighting, mesialization, distalization, or intrusion. In this study, 11 out of 36 miniscrews failed before the end of treatment, corresponding to a high failure rate of 30%. Again, the failure rate tended to decline with the time duration of the study and with increasing operator's experience or the ascending learning curve.

It is natural to expect that clinicians inexperienced in TAD placement are likely to have more potential failures since they are learning a new technique. Lim et al. [3] reported that practitioners who inserted more than 20 self-drilling miniscrews obtained approximately a 3.6-fold higher rate of initial stability compared to TADs implanted by less experienced clinicians. The authors considered "wobbling" or an inadequate miniscrew angle as the two major factors responsible for failure caused by an inexperienced operator. At this point, it is worth noting that although Lim et al. [3] achieved a high overall success rate of 93.1% (379/407), they assessed stability only one week after placement, while most miniscrews usually fail several weeks or months after placement [11,12]. In our study, a period of observation lasted 16 (±5) months; therefore, we avoided bias resulting from reporting "success" prematurely. Still raising the issue of comparison of experience and its lack, in the study by Chen et al. [6] there was no significant difference in the failure rates between an oral surgeon and an orthodontist who were both introduced to a new mini-implant system. They both exhibited the same learning curve regardless of the fact that the oral surgeon had inserted hundreds of pre-drilling miniscrews in the past. However, the novel technique, namely, the self-drilling type of miniscrews, ensured the oral surgeon encounter had a high failure rate, and again, such a result confirms the necessity of the gradually progressing learning curve in order to obtain excellence in any orthodontic or dental field. Furthermore, the same oral surgeon [13] lost 17.4% of TADs as a novice in inserting pre-drilling miniscrews, and then—three years later—reduced that number to 9.5% [6].

A lack of experience with TAD placement may possibly result in TAD insertion in the proximity of dental roots, which is recognized as a major factor for TAD loss [14–16]. In a study by Cho et al. [17], experienced and inexperienced dentists drilled 192 and 240 holes in phantoms, respectively. It turned out that novice operators generated a statistically significant

higher occurrence of root contacts (21.3%) than their advanced colleagues (13.5%). Similar results were reported in the in vitro study by Pérez et al. [18], where an orthodontist student without experience inserted stainless steel alloy orthodontic miniscrews less accurately than an orthodontist with 10 years of experience, which resulted in more intraoperative complications (root perforations). Therefore, it is no wonder that even fear of complications is a psychological factor restraining inexperienced doctors from performing new surgical procedures [19]. Antoszewska et al. [15] evaluated this issue in a group of 35 orthodontists placing miniscrews for the first time in typodonts with the gingival part covered with an opaque tape. The fear level was measured before and after the experiment and ranked on a 10-point visual scale. Regardless of the fact that the percentage of TADs that were in contact with the dental root was relatively high (23.57%), the mean fear level declined from 4.6 before the insertion to 3.2 after this procedure. The risk of the root, maxillary sinus, or mandibular canal injury occupied the main positions on the list of scaring factors. It can therefore be concluded that providing detailed theoretical and practical training to orthodontists may diminish their fear level. In this context, the particularly important result of our study is the one proving that with every procedure, the probability of the miniscrew success increased by 1.016 times, and it was regardless of whether the operator was a titled scientist (JA-S, the associate professor, and MS, the lecturer) or a newcomer (JŁ, the postgraduate student).

## 5. Conclusions

1. Just like any other technique, TAD placement has a learning curve.

2. Decent survival rates of 80% and 85% were established after the insertion of 74 and 118 TADs, respectively. Moreover, the success rate significantly increased (1.016 times) after each procedure of TAD insertion in the buccal maxillary area. These findings might encourage hesitating clinicians to introduce the miniscrews to their daily practices, especially for en masse retraction or the distalization of maxillary teeth.

3. The opportunity to learn from more experienced colleagues could be both beneficial and productive for novice clinicians. Thus, it seems advisable to create an effective training system that emphasizes all of the factors related to TAD stability prior to their placement.

**Author Contributions:** Conceptualization, M.S. and J.L.; methodology, M.S.; software, M.S.; validation, J.L., M.S., and B.K.; formal analysis, M.S. and K.R.; investigation, J.L., M.S., and K.R.; resources, K.R.; data curation, J.L., M.S., and K.R.; writing—original draft preparation, K.R. and J.L.; writing—review and editing, B.K.; and supervision, B.K. All authors have read and agreed to the published version of the manuscript.

**Funding:** This research received no external funding.

**Institutional Review Board Statement:** The study was conducted according to the guidelines of the Declaration of Helsinki.

**Informed Consent Statement:** Informed consent was obtained from all subjects involved in the study.

**Data Availability Statement:** The data presented in this study are available on request from the corresponding author.

**Conflicts of Interest:** The authors declare no conflict of interest.

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
