# Peer review of "Effect of the Increasing Operator’s Experience on the Miniscrew Survival Rate"

_applsci, doi:10.3390/app122211647_

Round 1
Reviewer 1 Report
This report is examining the effect of the operator's experience in miniscrew implantation. It is an interesting report but some parts should be revised.
1) In the introduction section, you stated what the null hypotheses are. However, you should state the hypotheses before stating the null hypothesis.
2) In addition, you should add a statement about what your hypothesis resulted in.
3) Figure 1 is not raw data. Fig 1 and Fig 3 should be combined into a Table for clarity.
4) Table 4: About explanatory variable. What does “Patient” mean?
5) In addition, please add its explanation into the text, and please add the discussion about the meaning of “1.016 times” of odds ratio in detail.
6) In the discussion section, the first 2 lines should be removed.
Reviewer 2 Report
Dear authors,
I advise you to change following:
Introduction:
change: … finding should definitely encourage hesitating… to more moderate expression as …might…;
also change the flow of the text so the whole expression (i.e. odds ratio = 1.016,) stays whole on one row.
Material and Methods:
… All patients presented skeletal Class II, with either hypodivergent or normal angle between the maxillary and mandibular planes, with excellent oral hygiene… Please specify, how was the level of dental hygiene determined.
In „statistical analysis“ change …was set at p = .05… to …p = 0.05… as it is used in the table 1.
Fig. 1
Change the image captation to more detailed one – „all operators raw data“ is not enough to understand the image.
TADs implantation might be not the best expression as in the rest of the article more fitting world „insertion“ is used.
Change …Probability of succes… for …failure/success ratio…mentioned in the results. Or describe the term probability of succes in the text.
Fig. 2
TADs implantation might be not the best expression as in the rest of the article more fitting world „insertion“ is used.
Change …Probability of succes… for …failure/success ratio…mentioned in the results. Or describe the term probability of succes in the text.
Fig.3
Change the image captation to more detailed one – „each operators raw data“ is not enough to understand the image.
TADs implantation might be not the best expression as in the rest of the article more fitting world „insertion“ is used.
Change …Probability of succes… for …failure/success ratio…mentioned in the results. Or describe the term probability of succes in the text.
Fig. 4
TADs implantation might be not the best expression as in the rest of the article more fitting world „insertion“ is used.
Change …Probability of succes… for …failure/success ratio…mentioned in the results. Or describe the term probability of succes in the text.
Results
In „statistical analysis“ change …was set at p = .006… to …p = 0.006… as it is used in the table 1.
Discussion:
Delete the sentence: Since we could not find any paper that presented exact same design as ours, the discussion below is based on a limited number of studies. as it is unnecessary.
Word …Unquestionably…also the expression …But back to the experience issue… are not necessary in the text, please consider deleting them.
No personal skills of any clinician were mentioned in the discussion. It might be important and influence results and differences between the three clinicians.
Line 155 – TADs are described as microimplants (as in the original study) however it might lead to confusion of the reader, consider to unify the terms with the rest of the text.
Line 192 – dash before 192 and 240 is probably an error
Last sentence :… And it was regardless of whether the operator was a titled scientist or a newcomer…. Please describe in other words or more clear terms.
Conclusions:
Line 219 – remove space before the word “each”.
change: … finding should definitely encourage hesitating… to more moderate expression as …might…;

Round 2
Reviewer 1 Report
Thank you for conscientiously addressing the original manuscript.